# Two Gold Kiwifruit Daily for Effective Treatment of Constipation in Adults—A Randomized Clinical Trial

**DOI:** 10.3390/nu14194146

**Published:** 2022-10-06

**Authors:** Simone B. Bayer, Phoebe Heenan, Chris Frampton, Catherine L. Wall, Lynley N. Drummond, Nicole C. Roy, Richard B. Gearry

**Affiliations:** 1Gastrointestinal Unit for Translational Studies, Department of Medicine, University of Otago Christchurch, 2 Riccarton Avenue, Christchurch 8011, New Zealand; 2New Zealand National Science Challenge—High-Value Nutrition, Liggins Institute, University of Auckland, Building 505, 85 Park Road, Auckland 1023, New Zealand; 3Biostatistics and Computational Biology Unit, University of Otago Christchurch, 2 Riccarton Avenue, Christchurch 8011, New Zealand; 4Drummond Food Science Advisory Ltd., 1137 Drain Road, Killinchy 7682, New Zealand; 5Department of Human Nutrition, University of Otago, 7n8, Level 7, Science 2 Building, 70 Union Street West, Dunedin 9016, New Zealand

**Keywords:** constipation, gastrointestinal symptom rating scale, irritable bowel syndrome, kiwifruit, psyllium, straining

## Abstract

Chronic constipation is highly prevalent worldwide and may be managed with two green or three gold kiwifruit daily. It is unknown whether a smaller standard serve of gold kiwifruit (two daily) is as effective in constipation management. The study aimed to improve chronic constipation with two gold kiwifruit and psyllium in lieu of a placebo daily over four weeks. Adult participants (18–65 years) with functional constipation (FC, *n* = 11), constipation-predominant irritable bowel syndrome (IBS-C, *n* = 13), and healthy controls (*n* = 32) were block-randomized to the treatment order: gold kiwifruit (2/day) or psyllium (fiber-matched, 7.5 g/day) for four weeks, followed by four weeks washout before crossover. Outcomes included alterations of Gastrointestinal Symptom Rating Scale (GSRS) domains and weekly complete spontaneous bowel movements (CSBM) as part of a larger study. Both interventions reduced GSRS constipation domain scores in all subjects compared to baseline values (*p* = 0.004). All participants reported significantly more weekly CSBM (*p* = 0.014). Two gold kiwifruit decreased straining (*p* = 0.021). Two gold kiwifruit daily are as effective as fiber-matched psyllium in treating constipation in adults and should be considered as a treatment option.

## 1. Introduction

Up to 60% of the global population report symptoms of chronic gastrointestinal discomfort such as chronic constipation [1,2]. Rome IV criteria define chronic constipation as part of the functional bowel disorder continuum with functional constipation (FC) and irritable bowel syndrome with constipation (IBS-C) [1,3], since FC and IBS-C symptoms overlap [4]. There are increased health care costs associated with IBS [5,6,7], a decreased quality of life [8], and increased co-morbidities such as anxiety and depression [9,10].

The mechanisms of IBS-C and FC pathogenesis are multifactorial [11,12], with significant symptom variability between and within patients over time [3]. Common treatments for IBS-C include pharmacotherapy, dietary and lifestyle changes [12]. Due to the multifactorial pathogenesis, available treatments are mostly unsatisfactory and only address symptoms. Because of the chronic and prevalent nature of IBS-C and FC, the demand for accessible, effective treatment is high. Dietary changes are thought to be effective treatments for IBS-C [13,14] but are rarely studied [15].

One of the foods studied in clinical trials for the treatment of chronic constipation is green kiwifruit *Actinidia deliciosa* ‘Hayward’. More recently, green kiwifruit has been recognized as a food able to manage the symptoms of constipation in clinical studies [16,17] and by the European Food Safety Authority [18]. Strong evidence of laxation, relief from discomfort, and positive effects on gut microbiota composition have been consistently reported [11], and the consumption of two green kiwifruit improved symptoms of digestive function in participants with FC [17,19]. Similar beneficial effects on laxation have been shown in constipated participants consuming three gold kiwifruit *Actinidia chinensis* ‘Zesy002’ (SunGold*™*) per day over 28 days [20]. A standard serve of kiwifruit consists of two medium sized fruit [21]. Furthermore, while both kiwifruit cultivars are nutrient-rich, and have a similar fiber composition [22,23,24], gold kiwifruit has a higher vitamin C content, and the proteolytic activity of the kiwifruit enzyme actinidin is eight times higher in green kiwifruit [25]. This is of particular interest because the original gold kiwifruit *Actinidia chinensis* ‘Hort16’ had no actinidin activity and was ineffective as treatment for the relief of gastrointestinal discomfort (unpublished results, L. N. Drummond). Whether a more acceptable standard serve of two gold kiwifruit is as effective remains to be determined.

As a complex biological material, a whole fresh fruit, gold kiwifruit cannot be easily adequately matched by a simple control material for a placebo. The fiber content of kiwifruit is believed to be a key contributor to the efficacy of kiwifruit in addressing digestive comfort [11]. On this basis, psyllium (Ispaghula), a well-known soluble fiber supplement and a well-studied, well-tolerated therapeutic for multiple functional gastrointestinal disorders (FGID) [12,16,17,18], was dose-matched for fiber content in lieu of a true placebo. The use of fiber-matched psyllium as a control intervention has been accepted as appropriate [20,26].

The study hypothesized that the consumption of two gold kiwifruit per day over 28 days is similarly effective to fiber-matched psyllium in the management of constipation.

The previously developed “Christchurch IBS cohort to investigate mechanisms for gut relief and improved transit” (COMFORT), an observational study using Patient-Reported Outcomes (PRO) related to symptoms, dietary records, and a multi-omics approach to assess biological markers [27], was adapted into a clinical trial. The COMFORT methodology entails information by PRO such as digestive health and mood, and multi-omics analyses (metabolome and microbiome) of biological samples (breath, blood, urine, and feces), and dietary data. The stated primary outcome was to determine whether the comprehensive assessments undertaken in the COMFORT cohort could be translated into a clinical trial. However, during recruitment, the design was found to be acceptable to participants, so recruitment was expanded to provide more definitive outcome data on efficacy and safety. Independently, the main clinical outcome was an improvement of the Gastrointestinal Symptom Rating Scale (GSRS) constipation domain score with both interventions [28]. Other secondary outcomes were increased weekly Complete Spontaneous Bowel Movements (CSBM) by one unit.

The data presented here is the first report on the results of COMFORT-PSYKI, focused on clinical outcomes.

## 2. Materials and Methods

This clinical trial was a single-blinded, randomized crossover design. Patients were block-randomized (block size 4) to the order of intervention, with a washout period before crossover. The protocol was approved by the New Zealand Human Disability and Ethics committee (18/STH/154) and prospectively registered with the Australian New Zealand Clinical Trial Registry (ACTRN, 1261 8001 2862 35p).

Participants were recruited through the COMFORT database, advertisements on local newsletters, websites, posters, and social media presence. Subjects between the ages of 18 and 65 years with a BMI between 18 and 35 kg/m^2^ were eligible.

Participants were included in the FC and IBS-C groups according to Rome IV criteria [3], and in the healthy control group in absence of gastrointestinal symptoms.

Participants were excluded if they were unable to consent, had a known kiwifruit or latex allergy, blood glucose of ≥6.0 mmol/L, were unable to stop laxative use for the week before sampling, IBS Symptom Severity Index (IBS-SSI) of over 300 [29], were pregnant, breastfeeding or planning pregnancy during the trial, had alarm features associated with bowel habits, family history of GI cancer, known significant GI disorder other than IBS-C, presence of severe chronic disease, previous GI surgery, or neurological disorders.

### 2.1. Study Protocol

Study volunteers were invited to participate in a study to evaluate the effect of two gold kiwifruit daily on constipation symptoms. Researchers responsible for analysis were blinded to the intervention order, while unblinded study personnel organized randomization and intervention distribution. After informed consent was given, potential participants underwent comprehensive clinical assessment, including biochemical blood analysis and collection of anthropometric data, during screening. Participants’ gastrointestinal symptoms were evaluated for severity and classification as FC, IBS-C, or healthy controls using Rome IV criteria [3]. Eligible participants were enrolled and instructed to discontinue laxatives for at least the week before biological sample collections (except for the rescue medication bisacodyl suppositories) and avoid fiber supplements and kiwifruit other than supplied, while otherwise maintaining their lifestyle and diet. After a 2-week lead-in period, the participants were randomized and supplied either two gold kiwifruit *(Actinidia chinensis* SunGold^TM^, Zespri^TM^ International Ltd., Mount Maunganui, New Zealand, fiber ~2.5 g) or 1.5 level teaspoons of psyllium (BonVit^®^ Orange, BonVit^®^, Castle Hill, Australia, fiber ~2.5 g) per day for four weeks (Table 1). As detailed in the introduction, the dose of psyllium is matched to the dietary fiber concentration present in kiwifruit after food waste adjustment (skins) in lieu of a placebo and is therefore below the recommended therapeutic dose. A four-week washout period followed before crossover into another four-week intervention period to receive the alternate intervention, then a two-week follow-up period (Figure 1).

Participants were allocated into blocks of four and randomized in equal numbers by pulling numbers out of a container to the order of intervention.

Eligible participants met with the unblinded researcher after the lead-in period to discuss the treatments. Participants were instructed to consume one and a half level teaspoons of psyllium a day, dissolved in liquid, or two peeled kiwifruit daily, unblended, and unheated, depending on intervention order, for four successive weeks. Participants were instructed to complete a three-day Food and Symptoms Times Diary [30]. The completed diet diaries were collected after each intervention period and at the end of the washout period (Figure 1). A rescue laxative (bisacodyl suppositories) was available upon request the week before a sample collection appointment; otherwise, participants could take laxatives.

### 2.2. Symptom Assessment

Over 16 weeks, participants were asked to complete a daily bowel movement (BM) diary, and a weekly survey on gastrointestinal symptoms and mental health. The weekly survey consisted of the GSRS (Astra Zeneca) [28]; the Structured Assessment of Gastrointestinal Symptoms (SAGIS, Universities of Queensland and Newcastle, Appendix A, SAGIS and PROMIS results) [31], and modules of the Patient-Reported Outcomes Measurement Information System (PROMIS, Health Measures, Appendix A, SAGIS and PROMIS results) [32,33]. Surveys were available in electronic form or printed versions for participants.

### 2.3. Clinical Endpoints

The independent, main clinical outcome was a change in the constipation domain score of the GSRS with both interventions. Each of the 15 GSRS items is rated through a 7-point Likert-scale. Items cover five domains: diarrhea, abdominal pain, indigestion, constipation, and reflux. An average score is calculated for each domain after completion [28]. Other secondary endpoints were an increase of ≥1 CSBM weekly with both interventions, improvements in straining, and stool consistency according to the Bristol Stool Form Scale (BSFS) [34]. A CSBM is a complete evacuation not induced by laxatives, rescue medication, or enemas. Straining, laxative use and manual evacuation techniques were determined using a yes/no question in addition to the BSFS in the Daily BM diary.

### 2.4. Protocol Deviation

Minor protocol deviations were the inclusion of two participants with fasting blood glucose >6 mmol/L after approval from the medical primary investigator, and three cases of phase extension by one week after mild digestive upset. A major protocol deviation was the premature end of breath sample collection due to the inability to procure additional collection tubes. None of the deviations had any impact on study integrity or data analyses.

### 2.5. Statistical Analysis

To compare baseline characteristics and demographics between diagnostic groups, χ2 tests, one way ANOVA and Kruskal–Wallis tests were used as appropriate.

Data from GSRS were treated as continuous data. For CSBM, BSFS, and straining, the analysis was based on the number of BMs. During the last week, the scores of each intervention were subtracted from the scores collected in the week before each intervention (baseline week 2, and washout week 4), resulting in a delta score (Δ). The Δ-scores were analyzed using ANOVA to compare changes between participant groups and are presented as least square means and 95% confidence intervals. The sequence was included as a between-subjects factor in the analysis to compare interventions.

A two-tailed *p* ≤ 0.05 was taken to indicate statistical significance, and there was no adjustment for multiple comparisons. Incomplete data were not imputed.

The sample size was calculated based on GSRS data from a similar study using three gold kiwifruit [20]. Statistical significance was set at *p* ≤ 0.05. Thirty participants were needed per group to detect an effect size of 1 Likert-score within the GSRS constipation domain with 5% α-level and 80% power for a difference between groups, and an effect size of 0.6 in all participants between interventions. With an expected dropout rate of 20 participants, the aim was to recruit 40 participants for each group. IBM^®^ SPSS^®^ statistics version 25 (Armonk, New York, US) was used for analysis.

The patient-reported outcome data were also compared to the Minimal Important Difference (MID). MID is the smallest difference in a PRO score perceived as important by the participant or patient [35,36]. The MID can be estimated as 0.5 in a 7-item Likert scale PRO [35], although this can vary for different questionnaire domains (0.4–0.8) [37]. In this study, we compared the data of the GSRS to the existing MID. No MID has been calculated for this study, as it was outside of scope. Since MID is used to describe Likert-like scales, no MID exists for items in the Daily BM diary.

## 3. Results

### 3.1. Subject Characteristics

Of 101 participants that were invited for screening between April and August 2019, 73 participants (37 controls, 17 FC, 19 IBS-C) were enrolled and randomized, and 56 (77%) completed the study (Figure 2) by 23 December 2019. Seventeen participants were withdrawn (5 healthy controls, 6 FC, 6 IBS-C) for reasons shown in Figure 2. Two participants were excluded from the GSRS full-dataset analysis due to incomplete data. Demographic, ethnicity, anthropometry, and blood biochemistry results were comparable between groups, except IBS-SSI scores, which were significantly higher in FC and IBS-C (*p* < 0.0001, Table 2). Food diary data did not indicate any significant dietary alterations during the study other than interventions used.

### 3.2. Primary Clinical Outcome—GSRS Constipation Domain Scores

In cross over design studies, the baseline for each intervention is different depending on whether the intervention is first or second. Therefore, the data are presented as a relative shift from baseline. For GSRS, a reduction from baseline corresponds with an improvement in symptoms.

Both interventions significantly improved GSRS constipation scores compared to baseline values (Effect of interventions, *p* = 0.004, Figure 3, Table 3). Gold kiwifruit reduced mean Δ-scores by −1.08 (−0.60, −1.57) in FC and by −0.03 (0.48, −0.54) in IBS, while psyllium reduced the Δ-scores by −0.44 (0.05, −0.94) in FC and −0.45 (0.06, −0.97) in IBS.

The relative shift from baseline was significantly different between the participant groups (Effects of participant group, *p* = 0.044): controls experienced the least reduction in constipation symptoms, followed by IBS and FC, with both interventions. There was no statistical significant difference between gold kiwifruit and psyllium (Gold kiwifruit vs. psyllium, *p* = 0.407).

### 3.3. Secondary Outcomes

#### 3.3.1. CSBM and Laxative Use

With either intervention, the mean number of weekly CSBM increased significantly in participants with FC or IBS-C compared to baseline values (*p* = 0.014, Figure 4, Table 4). The mean weekly CSBM increase was numerically higher with psyllium 1.63 ± 0.98 CSBM for IBS-C and 1.57 ± 1.01 CSBM for FC than with kiwifruit at 0.99 ± 0.93 CSBM for IBS-C and 1.08 ± 0.98 CSBM for FC. There was no statistical difference between kiwifruit and psyllium. The observed weekly CSBM increase was accompanied solely by higher Complete BM, not by Spontaneous BM (Table 4). Manual evacuation techniques were low during the study (5.4% of Total BM).

Laxative use was negligible. Of 56 participants who completed the study, four (7.1%) used laxatives during the study, one participant more than once. Laxatives were used seven times during the study, independent of phase. No participant used rescue medication.

#### 3.3.2. Ease of Defecation

Psyllium and kiwifruit softened stool consistency significantly compared to baseline values (*p* = 0.027, Table 4). The mean BSFS score increased by 0.4 ± 0.3 with psyllium and 0.2 ± 0.2 with kiwifruit; the overall difference between the interventions and the participant groups was not significant. There were no statistically significant improvements in mean straining scores with either intervention. However, there was a significant difference between the effects of the two interventions on straining scores: gold kiwifruit reduced weekly straining scores in FC by an average of 1.1 ± 0.7 and in IBS-C by an average of 1.4 ± 0.7 occasions, while psyllium did not affect straining scores (0.8 ± 0.8 and 0.3 ± 0.7 occasions per week in FC and IBS-C, respectively, *p* = 0.021, Figure 5, Table 4).

#### 3.3.3. Adverse Events

Both interventions were generally well tolerated. However, 29 participants reported 43 adverse events (AE), with 25 likely being intervention related (18 psyllium, 7 kiwifruit, Appendix A, Adverse events). With psyllium, participants were more likely to report intervention-related AE (*p* = 0.02). Bloating (*n* = 7) was the most commonly reported AE. One intervention-unrelated serious AE was recorded; the participant was withdrawn and has fully recovered (Figure 2).

## 4. Discussion

This feasibility study is the first to determine the effectiveness of a standard serving of two gold kiwifruit daily over 28 days compared to psyllium in constipation management in a clinical trial. Both interventions improved GSRS constipation scores and increased CSBM by approximately one per week, and equally improved ease of defecation in constipated adults without difference between interventions. Gold kiwifruit appeared to improve straining more than psyllium. Both interventions did not affect any measured PRO in healthy controls.

The results in this study show that a lower, more acceptable and affordable standard serving of two gold kiwifruit daily still elicits similar outcomes as previous studies exploring the effects of two green [26] or three gold kiwifruit [20] in constipated adults. Due to the higher fructose content [38], gold kiwifruit may be more palatable for some individuals, and the higher vitamin C content of gold kiwifruit content may provide additional health benefits [24].

Previous clinical trials on kiwifruit have used Rome II [17] or Rome III criteria [16]. The 2016 update to Rome IV introduced higher thresholds for abdominal pain in IBS-C, resulting in greater variation of IBS-SSI scores between IBS-C and FC [1,39] and improved separation between FC and IBS-C [39]. Now, IBS-C represents a more serious disorder [40]. This shift was observed in this study, and IBS-SSI scores were higher for IBS-C than FC. This increased separation between FC and IBS-C could explain the observed numerical disparity in GSRS constipation scores between the FC and IBS-C groups, where kiwifruit seemed to elicit a larger benefit in FC than in IBS-C. Yet, this was not confirmed statistically, where the participant group effect between the controls and the two constipated groups was statistically significant. Both psyllium and gold kiwifruit reduce constipation symptoms equally.

Since the change in constipation domain scores was greater than the MID of 0.6, this improvement is clinically significant. This could have been caused by a shift of bowel habits from constipation towards diarrhea by kiwifruit, common in IBS-C but excluded as a symptom for FC [3]. Yet, no increases in GSRS diarrhea scores were evident. Weekly CSBM increased with gold kiwifruit by 0.99 ± 0.93 CSBM for IBS-C and 1.08 ± 0.98 CSBM for FC, and with psyllium by 1.63 ± 0.98 CSBM for IBS-C and 1.57 ± 1.01 CSBM for FC participants. An increase in CSBM of over one per week is generally considered as clinically meaningful for individuals with FC [41]. No MID exists for bowel habit diaries. Furthermore, the increase in BSFS was only moderate, and a change of stool consistency from constipation (forms 1–2) to loose and watery stools (forms 6–7) would cover four categories. Thus, the increase in CSBM was not driven by an increase in diarrhea as defined by stool form [34].

Consumption of two gold kiwifruit daily reduced straining scores, while psyllium did not. A similar amelioration of straining and BSFS ratings has been observed with three gold kiwifruit [20]. While stool consistency improvements have been reported with green kiwifruit [26,42], straining or BSFS were either not measured [16] nor observed [17].

Participants reported significantly less adverse events on kiwifruit than on psyllium, such as bloating. This finding confirms that gold kiwifruit is a safe and well-tolerated intervention, similar to green kiwifruit [16,17,20].

Strengths of this study encompass the single-blinded, randomized crossover design, the inclusion of a healthy control group, pragmatic clinical outcomes, and the use of food items as interventions that reduce confounding effects from processing or preparation [43].

Shortcomings of this study include the use of a suboptimal dose of a commercially available and registered product as an active control in lieu of a placebo, the exclusion of participants unable to restrict laxative use during the week before sample collection and a predominantly female participant cohort, which tends to reduce the generalizability of results. Other limitations are the possible inclusion of participants with pelvic floor dysfunction, which would not be ameliorated with both interventions. The small sample size may have introduced type II errors to results, “complete-dataset” analysis instead of “per-protocol” or “intention-to-treat” analysis may have increased attrition bias, and a lack of blinding of participants due to the nature of the interventions are also limitations.

## 5. Conclusions

In conclusion, this study provided evidence that habitual consumption of two gold kiwifruit daily, a realistic standard serving of fruit, is at least as effective as fiber-matched psyllium in lieu of placebo in treating constipation in adults, with fewer side effects, and offers greater ease of defecation through the improvement of stool consistency and reduction of straining.

This study provides evidence addressing the need for increasing demand for accessible, effective treatment of chronic constipation through dietary changes to aid a desperate patient population already struggling with stigma [39]. In addition to the clinical symptom data reported here, this study will provide insights into the complex interplay of diet, gastrointestinal symptoms, resident microbiota, immune system, and central pain dysregulation from the analysis of the fecal microbiome and fecal/plasma metabolomes in the FC, IBS-C, and healthy control participants.

## Figures and Tables

**Figure 1 nutrients-14-04146-f001:**
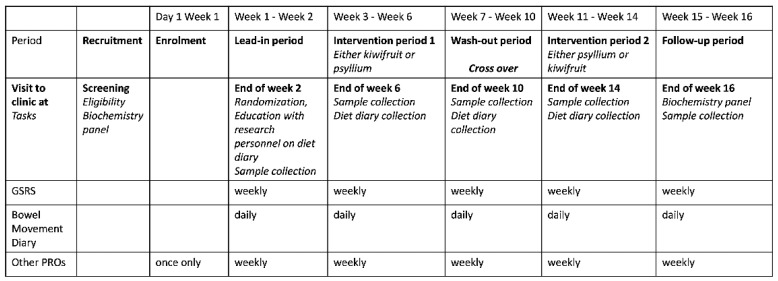
Study flow–diagram.

**Figure 2 nutrients-14-04146-f002:**
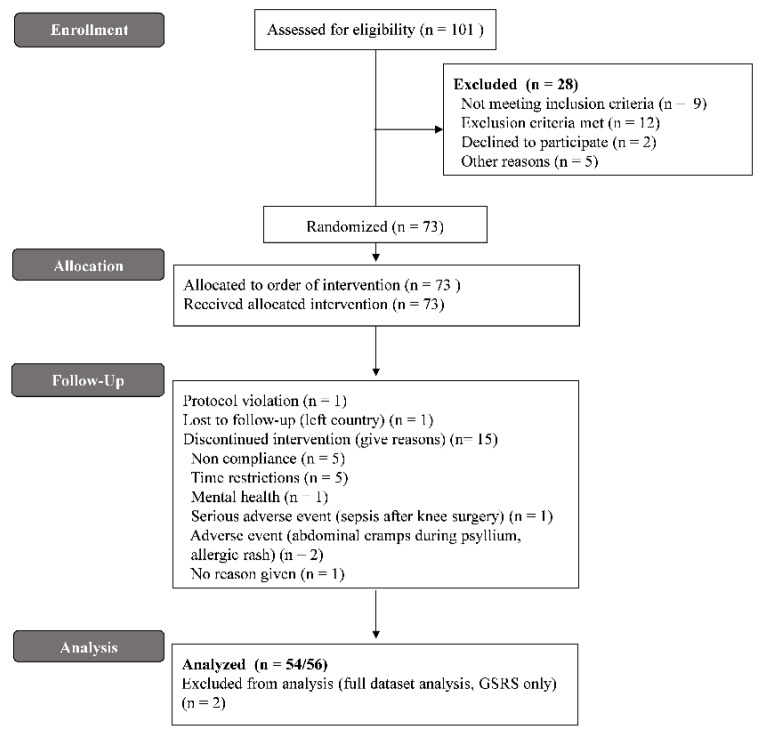
CONSORT flow diagram.

**Figure 3 nutrients-14-04146-f003:**
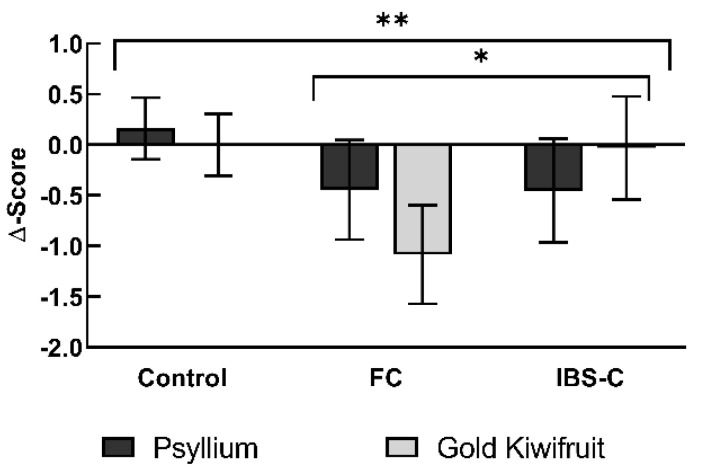
Gastrointestinal Symptom Rating Scale Constipation Domain Scores. Δ-score: median score after subtraction of pre-intervention score from post-intervention = relative shift from baseline. Bars: arithmetic means, error bars: 95% confidence intervals. Positive score = increase in symptom, negative score = decrease in symptom. *n* = 54. Asterisks indicate statistical significance. ** = Statistical significance as an effect of interventions, *p* = 0.004; * = Control vs. FC vs. IBS-C, Effects of participant group, *p* = 0.044. FC, Functional Constipation; IBS-C, Constipation-predominant Irritable Bowel Syndrome.

**Figure 4 nutrients-14-04146-f004:**
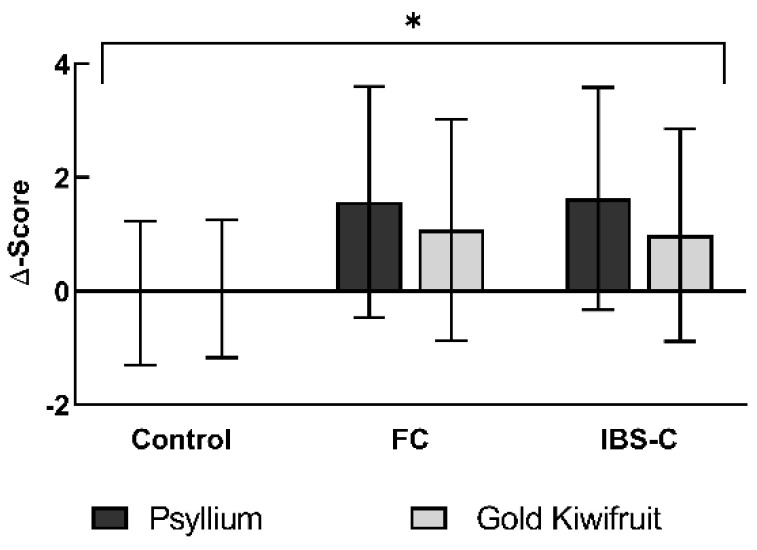
Complete Spontaneous Bowel Movements per week Δ-score: median score after subtraction of pre-intervention score from post-intervention = relative shift from baseline. Bars: arithmetic means, error bars: 95% confidence intervals. Positive score = increase in symptom, negative score = decrease in symptom. *n* = 56. Asterisks indicate statistical significance. * = Statistically significant effect of interventions, *p* = 0.006. FC, Functional Constipation; IBS-C, Constipation-predominant Irritable Bowel Syndrome.

**Figure 5 nutrients-14-04146-f005:**
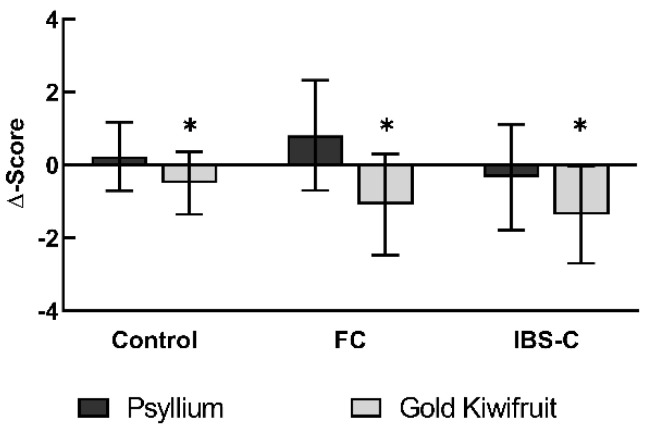
Changes in weekly straining scores. Δ-score: median score after subtraction of pre-intervention score from post-intervention = relative shift from baseline. Bars: arithmetic means, error bars: 95% confidence intervals. Positive score = increase in symptom, negative score = decrease in symptom. *n* = 56. Asterisks indicate statistical significance. * = Statistical significance between gold kiwifruit and psyllium, *p* = 0.021. FC, Functional Constipation: IBS-C, Constipation-predominant Irritable Bowel Syndrome.

**Table 1 nutrients-14-04146-t001:** Intervention composition summary.

	Gold Kiwifruit	Psyllium Preparation
Daily serving size	2 fruits, ~118 g per fruit	1.5 teaspoons, 7.5 g
Ingredients	SunGold kiwifruit	Husk powder, sugar, natural orange flavor/color, citric acid
Total fiber	1.4 g dietary fiber per 100 g ^1^	33.6 g dietary fiber per 100 g
Vitamin C	161.3 mg per 100 g fruit ^1^	nil
Sugar	12.3 g per 100 g fruit ^1^	40 g sucrose per 100 g ^2^

^1^ According to USDA food database, accessed on 23 November 2020. ^2^ BonVit^®^ Orange psyllium preparation, according to manufacturer, May 2019.

**Table 2 nutrients-14-04146-t002:** Baseline subject characteristics and IBS Symptom Severity Index (IBS-SSI).

Attribute	Control (*n* = 32) ^1^	FC (*n* = 11) ^1^	IBS-C (*n* = 13) ^1^
Gender:			
Female	26 (81%)	10 (91%)	12 (92%)
Male	6 (19%)	1 (9%)	1 (8%)
Ethnicity (self-reported):			
NZ European	27 (84%)	9 (82%)	9 (69%)
Māori	0 (0%)	0 (0%)	2 (15%)
Other ^2^	5 (16%)	2 (18%)	2 (15%)
Age (years)	35.0 ± 2.7	40.2 ± 4.1	44.2 ± 3.2
BMI (kg/m^2^)	23.9 ± 0.6	24.9 ± 1.3	24.8 ± 1.2
Weight (kg)	67.2 ± 1.8	68.7 ± 2.8	71.3 ± 3.2
Height (m)	1.68 ± 0.01	1.66 ± 0.02	1.70 ± 0.09
IBS-SSI *	25.7 ± 6.6	136.7 ± 21.6	252.9 ± 10.9

^1^ Values are presented as mean ± SEM or counts (percentage). ^2^ Other = Chinese, Hong Kong, Samoan, Korean, Fijian, Indian, Ethiopian, Middle-Eastern, Latin-American. * *p* < 0.0001. FC, functional constipation; IBS-C, irritable bowel syndrome constipation; IBS-SSI, IBS Symptom Severity Index; BMI, body mass index.

**Table 3 nutrients-14-04146-t003:** Differences in Gastrointestinal Symptom Rating Scale Scores post–pre-intervention.

					*p*-Value Factor (*n* = 54)	
GSRS-Domain ^2^ (MID [37])	Participant Group	Gold Kiwifruit ^1^	Psyllium ^1^	Effect of Interventions ^3^	Effects of Participant Group ^3^	Gold Kiwifruit vs. Psyllium ^3^
Constipation	Controls	0.00 (0.30, −0.30)	0.16 (0.47, −0.14)	***p* = 0.004 ****	***p* = 0.044 ***	*p* = 0.407
(±0.6)	FC	−1.08 (−0.60, −1.57)	−0.44 (0.05, −0.94)			
	IBS-C	−0.03 (0.48, −0.54)	−0.45 (0.06, −0.97)			
Diarrhea	Controls	0.18 (0.44, −0.07)	0.08 (0.31, −0.16)	*p* = 0.722	*p* = 0.679	*p* = 0.281
(±0.4)	FC	0.00 (0.41, −0.41)	−0.03 (0.35, −0.41)			
	IBS-C	0.15 (0.58, −0.28)	−0.24 (0.15, −0.64)			
Indigestion	Controls	−0.02 (0.20, −0.25)	0.04 (0.21, −0.13)	***p* = 0.008 ****	*p* = 0.457	*p* = 0.056
(±0.7)	FC	−0.54 (−0.18, −0.90)	−0.33 (−0.06, −0.61)			
	IBS-C	−0.27 (0.10, −0.65)	0.11 (0.40, −0.17)			
Pain	Controls	0.03 (0.34, −0.28)	0.00 (0.15, −0.15)	*p* = 0.290	*p* = 0.750	*p* = 0.581
(±0.6)	FC	−0.31 (0.19, −0.80)	−0.25 (−0.01, −0.49)			
	IBS-C	−0.09 (0.43, −0.61)	0.09 (0.34, −0.16)			
Reflux	Controls	−0.05 (0.17, −0.27)	0.03 (0.21, −0.15)	*p* = 0.071	*p* = 0.120	*p* = 0.261
(±0.8)	FC	−0.08 (0.27, −0.43)	−0.25 (0.04, −0.54)			
	IBS-C	−0.41 (−0.04, −0.77)	0.05 (0.35, −0.26)			

^1^ Values presented as Δ-score = median (upper bound, lower bound 95% confidence intervals) score after subtraction of pre-intervention score from the post-intervention score = relative shift from baseline. Bold data represent statistical significance. ^2^ Values scored using ordinal scales. Seven-point Likert scale, 1 to 7: “No discomfort at all”, “Minor discomfort”, “Mild discomfort”, “Moderate discomfort”, “Moderately severe discomfort”, “Severe discomfort” and “Very severe discomfort”. ^3^ Combined Effect of interventions on the relative shift from baseline. Effects of participant group on the relative shift from baseline. Kiwifruit vs. psyllium = differences between the interventions on their effect on the relative shift. Bold indicates statistical significance. * *p* < 0.05, ** *p* < 0.01. FC, Functional Constipation; IBS-C, constipation-predominant Irritable Bowel Syndrome; GSRS, Gastrointestinal Symptoms Rating Scale, MID, Minimal Important Difference.

**Table 4 nutrients-14-04146-t004:** Differences in Complete Daily Bowel Movement Diary results post–pre-intervention.

				*p*-Value Factor (*n* = 54)		
Daily BM Diary ^2^	Participant Group	Gold Kiwifruit ^1^	Psyllium ^1^	Effect of Interventions ^3^	Effects of Participant Group ^3^	Gold Kiwifruit vs. Psyllium ^3^
Total BMs	Controls	−0.22 (1.14, −1.58)	0.25 (1.45, −0.95)	*p* = 0.906	*p* = 0.747	*p* = 0.134
	FC	−0.81 (1.37, −2.99)	0.47 (2.40, −1.45)			
	IBS-C	−0.56 (1.53, −2.66)	1.13 (2.98, −0.72)			
Complete BM	Controls	−0.03 (1.19, −1.24)	−0.01 (1.28, −1.29)	***p* = 0.006 ****	*p* = 0.981	*p* = 0.820
	FC	1.36 (3.31, −0.60)	1.49 (3.55, −0.58)			
	IBS-C	1.44 (3.32, −0.44)	1.78 (3.77, −0.20)			
Spontaneous BM	Controls	−0.25 (1.10, −1.60)	0.45 (1.70, −0.80)	*p* = 0.962	*p* = 0.913	*p* = 0.154
	FC	−0.87 (1.29, −3.04)	0.39 (2.40, −1.62)			
	IBS-C	−0.56 (1.52, −2.65)	0.74 (2.67, −1.19)			
CSBM	Controls	0.04 (1.26, −1.17)	−0.04 (1.23, −1.30)	***p* = 0.014 ***	*p* = 0.886	*p* = 0.632
	FC	1.08 (3.03, −0.87)	1.57 (3.60, −0.47)			
	IBS-C	0.99 (2.86, −0.88)	1.63 (3.58, −0.33)			
Manual Evacuation	Controls	0.00 (0.26, −0.25)	0.17 (0.67, −0.32)	*p* = 0.725	*p* = 0.312	*p* = 0.359
Techniques	FC	0.25 (0.66, −0.16)	0.08 (0.89, −0.72)			
	IBS-C	0.21 (0.60, −0.19)	−0.51 (0.26, −1.28)			
Straining	Controls	−0.50 (0.36, −1.36)	0.23 (1.17, −0.71)	*p* = 0.151	*p* = 0.623	***p* = 0.021 ***
	FC	−1.10 (0.29, −2.48)	0.81 (2.32, −0.70)			
	IBS-C	−1.37 (−0.04, −2.70)	−0.34 (1.11, −1.79)			
BSFS	Controls	0.34 (0.75, −0.07)	0.08 (0.57, −0.40)	***p* = 0.027 ***	*p* = 0.643	*p* = 0.842
	FC	0.29 (0.95, −0.37)	0.46 (1.24, −0.32)			
	IBS-C	0.12 (0.75, −0.52)	0.37 (1.12, −0.38)			

^1^ Values presented as Δ-score = median (upper bound, lower bound 95% confidence intervals) score after subtraction of pre-intervention score from the post-intervention score = relative shift from baseline. Bold data represent statistical significance. ^2^ Values scored using actual number of BM, “Yes/No” for complete, spontaneous, laxative use, manual evacuation method and straining for every BM. BSFS was rated from 1 to 7: “Separate hard lumps, like nuts (hard to pass)”, “Sausage shaped but lumpy”, “Like a sausage but with cracks on its surface”, “Like a sausage or snake, smooth and soft”, “softs blobs with clear-cut edges (passed easily)”, “Fluffy pieces with ragged edges, a mushy stool” and “watery, no solid pieces”. ^3^ Combined effect of interventions on the relative shift from baseline. Effects of participant group on the relative shift from baseline. Kiwifruit vs. psyllium = differences between the interventions on their effect on the relative shift. Bold indicates statistical significance. * *p* < 0.05. BM, ** *p* < 0.01. Bowel Movement; CSBM, Complete Spontaneous Bowel Movement; BSFS, Bristol Stool Form Score.

## Data Availability

Data available upon request pending approval of a research proposal, statistical analysis plan, and execution of a data-sharing agreement. For information on how to request access to the data, please e-mail the corresponding author.

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
