# Peer review of "Two Gold Kiwifruit Daily for Effective Treatment of Constipation in Adults—A Randomized Clinical Trial"

_nutrients, 2022, doi:10.3390/nu14194146_

Round 1

Reviewer 1 Report

 The manuscript “Two gold kiwifruit daily for effective treatment of constipation 2 in adults – a randomized feasibility trial with a multi-omics approach”.

It is an interesting study, but rises several major concerns:

       The study is presented with the aim of answer a question “if the local New Zealand research team would be able to successfully execute a clinical trial with multi-omics methods. Therefore, the primary outcome was the feasibility”. So it is presented as a feasibility study on the possibility of “successfully execute a clinical trial with multi-omics methods”. Although, omics are several times mentioned in the manuscript, it is not clear their role in the study.

-           Although this is presented as a feasibility study, most results, discussion, and conclusion are presented as clinical results useful to promote an alternative therapeutic approach and not as a feasibility study.

-          I would suggest a manuscript revision and presentation of the study as a exploratory study on the effectiveness of this therapeutic approach.

Two minor concerns:

-          Generic and specific names should be written in italic

-          Tables should be more reader friendly.

Author Response

The manuscript “Two gold kiwifruit daily for effective treatment of constipation 2 in adults – a randomized feasibility trial with a multi-omics approach”.

It is an interesting study, but rises several major concerns:

       The study is presented with the aim of answer a question “if the local New Zealand research team would be able to successfully execute a clinical trial with multi-omics methods. Therefore, the primary outcome was the feasibility”. So it is presented as a feasibility study on the possibility of “successfully execute a clinical trial with multi-omics methods”. Although, omics are several times mentioned in the manuscript, it is not clear their role in the study.

1.The study is presented with the aim of answer a question “if the local New Zealand research team would be able to successfully execute a clinical trial with multi-omics methods. Therefore, the primary outcome was the feasibility”. So it is presented as a feasibility study on the possibility of “successfully execute a clinical trial with multi-omics methods”. Although, omics are several times mentioned in the manuscript, it is not clear their role in the study. 

R:We omited the omics from heading and reduced the use of “omics” in text.

We used “part of a larger study” instead on one occation, and added details on others.

2.Although this is presented as a feasibility study, most results, discussion, and conclusion are presented as clinical results useful to promote an alternative therapeutic approach and not as a feasibility study

I would suggest a manuscript revision and presentation of the study as a exploratory study on the effectiveness of this therapeutic approach.

R: We elaborated on this in the manuscript as followed:

“The previously developed “Christchurch IBS cohort to investigate mechanisms for gut relief and improved transit” (COMFORT), an observational study using Pa-tient-Reported Outcomes (PRO) related to symptoms, dietary records, and a multi-omics approach to assess biological markers, was adapted into a clinical trial. The COMFORT methodology entails information by PRO such as digestive health and mood, and multi-omics analyses (metabolome and microbiome) of biological samples (breath, blood, urine, and feces), and dietary data. The stated primary outcome was to determine whether the comprehensive assessments undertaken in the COMFORT co-hort could be translated to a clinical trial. However, during recruitment the design was found to be acceptable to participants, so recruitment was expanded to provide more definitive outcome data on efficacy and safety.”

3.Generic and specific names should be written in italic

R:Fixed. 

4.Tables should be more reader friendly.

R: We have simplified the tables.

Reviewer 2 Report

This is an interesting, well-designed study on a common problem that would benefit from more palatable solutions. I look forward to seeing the multiomics data. Below are my comments:

1. Regarding novelty, if 2 green kiwifruit are effective for treating constipation, why wouldn't one assume that 2 gold kiwifruit are effective, as well? Because they have less actinidin? Perhaps you could elaborate by describing how actinidin is thought to be helpful for constipation.

2. You should have given psyllium at the recommended dose, not matched it to the fiber content of kiwifruit, as the effect of kiwifruit is not just attributed its fiber content. Patients, dietitians, and physicians want to know if 2 gold kiwifruit are as good as a normal dose of psyllium, not a partial dose. It is especially important to avoid the appearance of stacking the deck in favor of kiwifruit, given the last author's connections with Zespri.

3. In table 2, were all participants really 170cm tall?

4. I was confused in the results section by your comparisons of pre- vs. post intervention, irrespective of intervention. Why I should be interested in the composite effect of both interventions? I am interested in knowing whether kiwifruit are effective and how they compare to psyllium.

5. I think lines 376-380 in the conclusion ("Authors should discuss the results...The findings and their implications should be discussed...") were editorial notes that were included by accident.

Author Response

This is an interesting, well-designed study on a common problem that would benefit from more palatable solutions. I look forward to seeing the multiomics data. Below are my comments:

1. Regarding novelty, if 2 green kiwifruit are effective for treating constipation, why wouldn't one assume that 2 gold kiwifruit are effective, as well? Because they have less actinidin? Perhaps you could elaborate by describing how actinidin is thought to be helpful for constipation.

R: We elaborated briefly in the introduction. The orgininal gold kiwifruit ‘Hort 16’ had no actinindin activity and was ineffective for the relief of gastrointestinal discomfort. (unpublished results, L.N.Drummond)

2. You should have given psyllium at the recommended dose, not matched it to the fiber content of kiwifruit, as the effect of kiwifruit is not just attributed its fiber content. Patients, dietitians, and physicians want to know if 2 gold kiwifruit are as good as a normal dose of psyllium, not a partial dose. It is especially important to avoid the appearance of stacking the deck in favor of kiwifruit, given the last author's connections with Zespri.

R: We revised the introductory paragraph to include a more detailed explanation for our choice. As a complex biological material, a whole fresh fruit, this cannot be easily adequately matched by a simple control material. The fiber content of kiwifriuit is believed to a key contributor to the efficacy of kiwifruit in adressing digestive comfort. On this basis, psyllium, a well known laxative aid, was dose-matched for fibre content in lieu of a placebo. The use of fiber-matched psyllium as a control intervention has been accepted as appropriate by other reviewers.

3. In table 2, were all participants really 170cm tall?

R: Yes, if rounded to one decimal place. I have added two decimal places to provide more detail. 

4. I was confused in the results section by your comparisons of pre- vs. post intervention, irrespective of intervention. Why I should be interested in the composite effect of both interventions? I am interested in knowing whether kiwifruit are effective and how they compare to psyllium.

R: We have simplified the table. The relative shift from baseline by intervention for each group is presented in the table as mean Δ-score and CI. The significant difference between the two treatments was relabeled as “gold kiwifruit vs psyllium”, and the significant reduction of symptoms was relabelled as “Effect of Interventions”. We have also reworded the results section on the GSRS constipation score to make the results more apparent.

5. I think lines 376-380 in the conclusion ("Authors should discuss the results...The findings and their implications should be discussed...") were editorial notes that were included by accident.

R: Removed.